# EVALUATING RANKING LOSS FUNCTIONS IN PERFORMANCE PREDICTOR FOR NAS

## ABSTRACT

Performance evaluation is a critical but compute-intensive procedure in neural architecture search (NAS). To alleviate evaluation costs, performance predictors have been widely adopted to predict architecture performance directly. Recent studies have introduced ranking loss functions into predictors to focus on the architecture rankings instead of absolute accuracy, thus enhancing the ranking ability of performance predictors. Despite the successful application of ranking loss functions, the lack of comprehensive measure metrics and different experimental configurations make a fair comparison among these loss functions a huge challenge. Additionally, some well-known ranking loss functions have not been thoroughly examined in the context of performance predictors. In this paper, we conduct the first study for 11 ranking loss functions containing the existing and the novel ones by comparing their effectiveness in performance predictors under various settings. We find that: (i) The choice of ranking loss function has a major influence on the performance of predictors; (ii) the quality of the architectures searched by the predictor-based NAS methods is closely correlated with the predictor's performance on top-centered rank metrics, rather than traditional metrics like Kendall Tau. We believe these results and insights can serve as recommendations for the optimal loss function to employ in predictors across various search spaces and experimental conditions.

## 1 INTRODUCTION

Neural architecture search (NAS) aims to automate the design of neural architectures (Elsken et al., 2019) and has been widely applied in various domains (Liu et al., 2018a; Pham et al., 2018; Ghiasi et al., 2019; Wang et al., 2020). One bottleneck of early NAS is the expensive performance evaluation stage, which involves training architectures from scratch (Zoph & Le, 2016; Real et al., 2017). To tackle this issue, performance predictors have been proposed to estimate the final performance of unseen architectures (Luo et al., 2018; Wen et al., 2020; Liu et al., 2021). Specifically, performance predictors can learn the mapping from architecture design to the corresponding performance using only a small number of trained architectures, thus greatly reducing computational costs. However, it remains challenging to accurately estimate the absolute performance of a given architecture at the budget of limited training samples (Xu et al., 2021). To overcome this challenge, recent advancements have proposed a paradigm shift from architecture performance prediction to architecture ranking prediction by replacing the widely-used mean square error (MSE) loss in predictors with pairwise and listwise ranking loss functions (Xu et al., 2021; Zheng et al., 2024; Hwang et al., 2024). This transition allows predictors to easily rank architectures in the correct order and thus facilitate NAS to discover the top-performing architectures.

Although ranking loss functions endow predictors with a better ranking ability, a fair comparison among them is still lacking currently. The results in prior works were mostly obtained under different experimental conditions where predictor types, search spaces, data splits, and hyperparameter tuning vary a lot from each other. Besides, only a few ranking losses have been applied to performance predictors previously because most losses were originally designed for information retrieval. Meanwhile, the relationship between the predictors' performance in rank-based metrics and their results in NAS experiments has not been well investigated.

This naturally raises the following questions: how do pointwise, pairwise and listwise ranking losses compare to each other under consistent conditions? Furthermore, could a ranking loss with excellent performance on ranked-based metrics indicate promising searching results for predictor-based NAS methods?

We answer the questions above by studying 11 ranking loss functions across five search spaces: NAS-Bench-101 (Ying et al., 2019), NAS-Bench-201 (Dong & Yang, 2019), TransNAS-Bench-101 (Duan et al., 2021), NAS-Bench-NLP (Klyuchnikov et al., 2022) and DARTS (Liu et al., 2018b). These ranking losses can be categorized into pointwise, pairwise, listwise, and weighted ones, with five of them used in previous predictors and six implemented newly by ourselves. To ensure a fair comparison between ranking losses, we employ a uniform predictor based on a graph convolution network (GCN) encoder in our experiments. The ranking ability of different ranking losses is assessed from three dimensions: training portion, test portion, and rank-based metrics. We adopt Kendall Tau (Kendall, 1938) and top-centered metrics such as Rel@K and N@K to test their overall ranking performance along with their performance in identifying top-ranked architectures. Then, we evaluate the ability of each ranking loss to facilitate two predictor-based NAS frameworks: predictor-guided random search (Bergstra & Bengio, 2012) and predictor-guided evolutionary search (Real et al., 2019). The relationship between the performance of ranking loss on rank-based metrics and the performance of predictor-based methods is clearly reflected in our experimental results.

Our findings indicate that the choice of ranking loss greatly affects the performance of predictors. For example, ListMLE (Xia et al., 2008) achieves a Kendall Tau of 0.55 for the Autoencoder task in TransNAS-Bench-101-Micro while MSE loss performs poorly with a score of 0.13. We also change the loss function in more complicated predictors for a more suitable one and find significant improvements in ranking performance. In addition, we find that a ranking loss performing well on top-centered metrics is more likely to aid predictor-based NAS in discovering promising architectures. In other words, we disprove the intuition that a predictor with high Kendall Tau is certain to help find well-performing architectures in NAS experiments. We show that a high Kendall Tau only indicates good overall ranking performance instead of a strong ability to recognize top-performing architectures. The latter matters more for NAS because predictor-based NAS methods mainly focus on discovering the optimal architecture in the search space.

We summarize our main contributions as follows:

- We provide the first comprehensive study for a series of pointwise, pairwise, listwise, and weighted ranking loss in performance predictors for NAS across various settings.
- We analyze our experimental results and find two key insights: (i) the importance of ranking loss in performance predictors; (ii) the correlation between performance on top-centered rank metrics and the quality of the architecture discovered by predictor-based NAS.
- We release a framework containing 11 ranking loss functions across 13 NAS tasks, which can serve as recommendations for predictors under different conditions. Note that six of them are adapted from applications in other domains.

## 2 RELATED WORK

**Performance Predictor for NAS**. Performance predictors have recently emerged to boost the evaluation efficacy of NAS. Predictors can estimate the final performance of unseen architectures directly, allowing NAS methods to navigate the search space more efficiently. To improve the accuracy and generalization of predictors, many studies focus on extracting diverse features from architectures by opting for complicated models(Wen et al., 2020; Lu et al., 2023), optimizing architecture encoding(Ning et al., 2020; Yan et al., 2021) and incorporating tailored self-supervised tasks(Jing et al., 2022; Zheng et al., 2024). However, the design of loss functions for performance predictors plays an equally significant role but is still rarely explored.

**Ranking Loss**. Ranking losses, which stem from learning to rank in the field of information retrieval, can be generally categorized into three types: pointwise, pairwise, and listwise (Liu et al., 2009). Pointwise methods treat the ranking problem as a regression task and directly predict the exact relevance degree of each document. Pointwise ranking loss is optimized by minimizing the regret such as the square error (Cossock & Zhang, 2006) on the training data. Pairwise methods

are closer to the nature of 'ranking'. They focus on the relative ranking for a pair of documents and predict which one is preferred within each data pair. Aiming to reduce the number of incorrectly ranked pairs as much as possible, many pairwise methods are proposed based on support vector machines (Herbrich et al., 2000; Joachims, 2002), boosting Freund et al. (2003), neural networks (Burges et al., 2005), and other machine learning models (Tsai et al., 2007; Zheng et al., 2007). Also, MP loss (Cortes et al., 2007) is proposed to keep the magnitude of the labeled documents when ranking the document pairs. Listwise methods consider the entire set of documents and ranked output a list that is most likely to be the ground truth. They can be roughly divided into two sub-categories: measure-specific (Taylor et al., 2008; Chakrabarti et al., 2008) and measure-irrelevant (Xia et al., 2008; Cao et al., 2007) loss functions. Additionally, some weighted ranking losses (Burges et al., 2006; Weston et al., 2011; Wang et al., 2018) are proposed to better recognize the top-rank documents. Despite the success of the ranking loss, it was originally designed for information retrieval or other fields. The large difference between document label and architecture label makes it nontrivial to adapt the ranking loss to the architecture ranking prediction task. To address this, we implement several well-known ranking losses tailored for performance predictors.

**Intersection Between Ranking Loss and Predictors**. Most existing performance predictors primarily utilize MSE loss, which belongs to pointwise ranking methods (Luo et al., 2018; Liu et al., 2021; Lu et al., 2023). However, predicting the relative ranking between architectures is more critical and effective than estimating the absolute accuracy of architectures (Xu et al., 2021). Recent predictors have introduced pairwise and listwise ranking loss functions to enhance their ranking ability. ReNAS (Xu et al., 2021) and FlowerFormer (Hwang et al., 2024) separately employ the hinge ranking loss and the margin ranking loss to preserve the rankings between different architectures. DCLP (Zheng et al., 2024) adopts ListMLE (Xia et al., 2008), a listwise method, to achieve better overall ranking performance. Although ranking loss enables the predictors to rank architectures better, no prior works conducted a comprehensive and fair comparison between them under consistent settings to the best of our knowledge.

## 3 BACKGROUND

In this section, we first introduce the basics of different categories of ranking loss functions. Then we discuss the evaluation metrics used to measure the ranking ability of performance predictors. The detailed descriptions for 11 ranking loss functions are included in Appendix A.1.

### 3.1 RANKING LOSS FUNCTIONS

In this paper, we compare four different categories of 11 ranking loss: pointwise ranking loss, pairwise ranking loss, listwise ranking loss and weighted ranking loss. Note that weighted ranking loss is mostly developed from pairwise and listwise ones. Formally, given $N$ architecture-performance pairs $\{(x_i, y_i)\}_{i=1}^N$, the predictor takes architectures as input and output predicted values $\{\hat{y}_i\}_{i=1}^N$.

**Pointwise Ranking Loss**. MSE loss, the traditional option for performance predictors, is chosen to represent the pointwise ranking loss in our experiments. For each input architecture $x_i$, the predictor aims to minimize the square loss $(y_i - \hat{y}_i)^2$ between predicted scores and ground truth. After that, the architecture ranking is calculated based on the predicted scores.

**Pairwise Ranking Loss**. Four pairwise ranking loss functions are integrated into performance predictors. They are hinge ranking (HR) loss, margin ranking (MR) loss, bayesian personalized ranking (BPR) loss (Rendle et al., 2009) and magnitude preserving (MP) loss (Cortes et al., 2007). For each input architecture pair $(x_i, x_j)$, the predictor is encouraged to predict relative ranking order correctly. The overall ranking is based on the massive relative ranking pairs. Among them, MP loss is special because it penalizes not only the mis-ranked pairs but also the correct ones if the magnitude of the prediction value is large.

**Listwise Ranking Loss**. ListNet (Cao et al., 2007) and ListMLE (Xia et al., 2008) losses are tested in our experiments. They optimize the entire architecture list $\{x_i\}_{i=1}^N$ to approach the correct ranking order and directly output a predicted ranking list for architectures.

**Weighted Ranking Loss**. Four weighted ranking loss functions are employed in our experiments which assign more weights to the top-ranked architectures when computed. We first develop the HR

loss for two weighted variants (short for WHR 1 and WHR 2) using different assignment manners. WHR 2 assigns more weights to the top-ranked architectures than WHR 1. Additionally, we introduce the well-known weighted approximate-rank pairwise (WARP) (Weston et al., 2011) loss and LambdaLoss (Wang et al., 2018) that assign weights in different manners.

## 3.2 RANK-BASED METRICS

In this paper, we employ the following rank-based metrics to evaluate the predictors' overall ranking performance and performance in identifying top-performing architectures.

**Kendall Tau** (Kendall, 1938) ($\tau$). $\tau$ can reflect the ordinal association between a predicted score and the absolute accuracy of architectures for each task. We have $\tau \in [-1, 1]$ and the ranking correlation is stronger if $\tau$ is closer to 1. We opt for Kendall Tau as the representative ranking correlation metric for the sake of its popularity in the field of performance predictors (Ning et al., 2020; Liu et al., 2021; Lu et al., 2023; Hwang et al., 2024).

**Weighted Kendall Tau** (Vigna, 2015) ($\tau_w$). Compared to vanilla $\tau$, $\tau_w$ assigns weights to architectures with top performance. There is a hyperbolic drop-off in architecture importance according to the descending order of accuracy. Predictors will earn a higher $\tau_w$ if the top-performing architectures are better ranked.

**N@K**. N@K denotes the true rank of the architecture that has the highest accuracy among the predicted top K ones. It is used to evaluate predictors because the relative rankings between architectures with poor performance is of little importance (Ning et al., 2020). If a promising architecture is predicted to top K, the performance in identifying top-performing architectures is good.

**Rel@K**. Distinguished from N@K, Rel@K is sensitive to the actual value of architecture. Rel@K computes the ratio of the accuracy of architecture $A_K$ to that of the best one $A_{max}$. We have Rel@K$\in (0, 1]$. In view of its success in other ranking problems (Li et al., 2021), this metric is also suitable for evaluating the effectiveness of the predictor in NAS because the majority of previous works only report the accuracy of the best architectures instead of the rank (Wen et al., 2020; Liu et al., 2021; Jing et al., 2022; Yi et al., 2023; Zheng et al., 2024).

## 4 EMPIRICAL EVALUATION OF RANKING LOSS FUNCTIONS

We divide our experiments into two parts: evaluating the performance of each ranking loss in a variety of rank-based metrics (Section 4.1) and evaluating the effectiveness of predictor-based NAS methods (Section 4.2). We start by describing the experimental settings.

**Experimental Settings**. The experiments are conducted in five popular search spaces: NAS-Bench-101 (Ying et al., 2019), NAS-Bench-201 (Dong & Yang, 2019), TransNAS-Bench-101 (Duan et al., 2021), NAS-Bench-NLP (Klyuchnikov et al., 2022) and DARTS (Liu et al., 2018b). Note that TransNAS-Bench-101 includes the macro and the micro search space. Most predictors are only evaluated on the latter while our experiments include the results on both. Detailed information about these search spaces will be elaborated in Section B.1. Although performance predictors can benefit from ranking loss functions, their effectiveness is heavily dependent on model selection, data scale and hyperparameter tuning. To achieve a fair comparison, we opt for a uniform performance predictor which is composed of a four-layer GCN encoder and a three-layer MLP in our experiments. The detailed hyperparameter configuration is included in Appendix B.2.

### 4.1 EVALUATING THE PERFORMANCE PREDICTORS

The predictor is trained with a certain number of architecture-performance pairs and predicts the architectures in the test set. We evaluate the ranking loss functions in 13 different tasks with respect to four rank-based metrics: Kendall Tau, weighted Kendall Tau, N@K and Rel@K. The ranking loss is compared based on three dimensions: training portion, test portion and rank-based metrics. We also apply them to representative predictors for performance improvements in rank-based metrics.

| | NB101-CF10 | TB101_MICRO-AUTO | TB101_MACRO-AUTO | TB101_MICRO-OBJECT | TB101-MACRO_OBJECT | TB101_MICRO-SCENE | TB101_MACRO-SCENE | TB101_MICRO_JIGSAW | TB101_MACRO_JIGSAW | NBNLP-PTB | NB201-CF10 | NB201-CF100 | NB201-IMGNT |
|---|---|---|---|---|---|---|---|---|---|---|---|---|---|
| MSE | 63.23 | 12.71 | -10.17 | 47.34 | 52.12 | 53.46 | 64.01 | 45.36 | 50.26 | 17.28 | 61.01 | 63.00 | 63.30 |
| BPR | 64.33 | 54.72 | 74.82 | 44.81 | 54.59 | 53.36 | 65.28 | 45.77 | 55.72 | 22.30 | 63.46 | 61.98 | 61.60 |
| HR | 64.37 | 54.73 | 74.73 | 44.95 | 54.30 | 53.37 | 65.11 | 45.74 | 55.48 | 22.32 | 63.43 | 62.91 | 61.75 |
| MR | 64.67 | 54.74 | 74.73 | 45.09 | 54.62 | 53.54 | 65.72 | 45.56 | 55.44 | 23.14 | 63.59 | 63.10 | 62.05 |
| MP | 64.30 | 26.41 | 51.36 | 47.60 | 51.36 | 53.59 | 63.15 | 45.30 | 49.06 | 17.95 | 60.11 | 62.46 | 63.23 |
| ListMLE | 63.79 | 55.14 | 74.24 | 45.17 | 52.99 | 53.53 | 64.20 | 46.10 | 54.44 | 23.16 | 63.99 | 62.77 | 62.22 |
| ListNet | 65.26 | -3.00 | 34.81 | 47.11 | 53.86 | 53.24 | 65.23 | 44.29 | 53.85 | 15.93 | 61.22 | 62.39 | 62.20 |
| WHR 1 | 64.52 | 54.87 | 75.02 | 44.66 | 54.97 | 53.64 | 65.77 | 45.93 | 56.59 | 22.32 | 63.36 | 62.79 | 61.54 |
| WHR 2 | 62.58 | 0.06 | 3.35 | 33.10 | 48.50 | 46.29 | 58.00 | 37.86 | 56.14 | 8.63 | 61.51 | 60.98 | 57.47 |
| WARP | 60.36 | 27.96 | 53.27 | 43.07 | 53.37 | 52.02 | 59.84 | 40.30 | 48.71 | 15.95 | 59.69 | 60.52 | 58.56 |
| LambdaLoss | 61.68 | 43.65 | 43.65 | 40.95 | 50.15 | 51.86 | 56.34 | 38.99 | 46.03 | 19.93 | 58.15 | 59.61 | 57.58 |

Figure 1: Kendall Tau of different ranking loss functions in 13 different tasks across four search spaces (scaled up by a factor of 100). The results are averaged over 100 trials.

### 4.1.1 PERFORMANCE IN RANKING CORRELATION

We first use Kendall Tau to evaluate the overall ranking performance of 11 ranking losses for 13 tasks across four search spaces. In this experiment, the training portion varies from 0.02% to 1.25% for different tasks according to the scale of the search space and the test portion is fixed to 100%. The results are reported in Figure 1. Each column represents the performance on a given task in a specific search space.

**Results on Different Tasks.** From Figure 1, we can observe that no ranking loss functions consistently surpass other competitors across all 13 tasks. For instance, ListNet achieves the top-1 $\tau$ in NAS-Bench-101 while having the lowest $\tau$ in the TransNAS-Bench101-Micro Autoencoder task. Additionally, most pairwise ranking loss functions such as BPR, HR and MR obtain promising performance in general. Even in the difficult TransNAS-Bench101-Macro Autoencoder task, they still attain a high $\tau$ of over 0.74, outperforming most of the other loss functions by a large margin. Conversely, the majority of weighted ranking loss functions like WHR 2, WARP and LambdaLoss perform poorly in Kendall Tau because they focus more on the ranking between top-performing architectures. WHR 1 is an exception because its weight assignment is not significant and preserves the ability of HR in overall architecture ranking.

To explore the effect of ranking loss in various predictor scenarios, we further compare them with respect to Kendall Tau using 16 combinations of training portion and test portion in different tasks in Figure 2. For each (training portion, test portion) pair, the optimal ranking loss and the corresponding $\tau$ are marked. Note that the ranking losses of the same category are plotted with similar colors for clarity. For example, pairwise ranking losses are colored with different types of blue.

**Results on Different Data Splits.** For page limit, the complete results of other tasks are reported in Appendix C.1. Figure 2 clearly illustrates that the optimal ranking loss under different data splits is not consistent even in the same task. In most tasks, only three or four out of the 11 ranking loss own the best performance over 16 data splits. Besides, the best ranking loss changes frequently and gets significant performance improvements in each row. But from each column, the optimal loss changes a little and the performance is stable. This indicates that the training portion is more significant to the optimal option of ranking loss in terms of $\tau$ for each task.

When the training portion is small, the completion for the optimal loss is fierce. At the budget of 0.5% training data on NAS-Bench-201, ListMLE beats others in CIFAR10 and CIFAR100 dataset (Krizhevsky et al., 2009) while MSE performs best in ImageNet16-120 dataset (Chrabaszcz et al., 2017). For the largest NAS-Bench-101, MP achieves the highest $\tau$. When it comes to TransNAS-Bench-101-Macro, WHR 1 is the most competitive both on the class-scene and class-

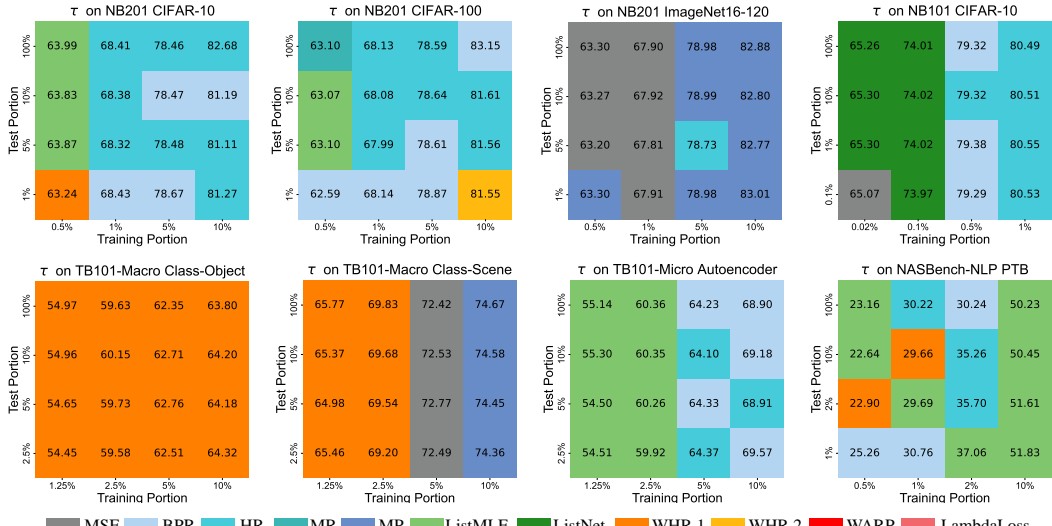

Figure 2: Kendall Tau of different ranking loss functions under various settings (scaled up by a factor of 100). The MSE loss is colored in gray. The pairwise ranking losses are colored in blue. The listwise ranking losses are colored in green. The weighted ranking losses are colored in red and orange. The results are averaged over 100 trials.

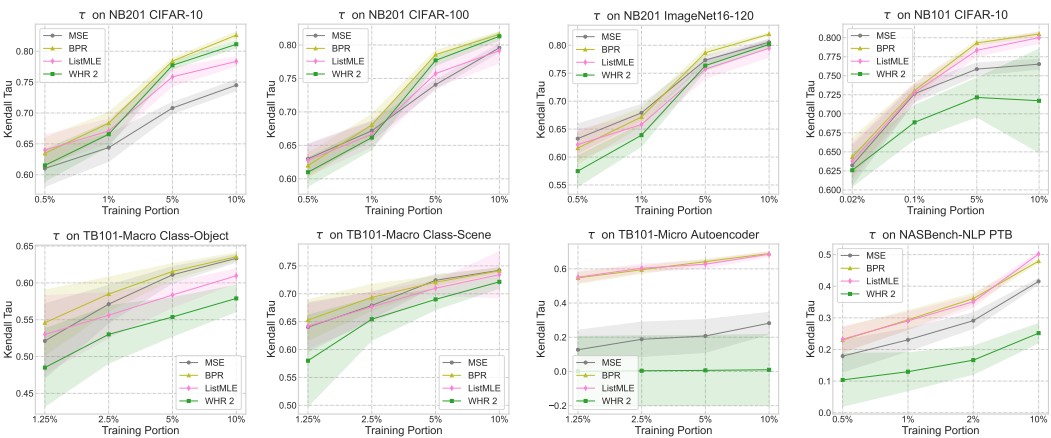

Figure 3: Kendall Tau of different ranking loss functions as the training portion grows. The results are averaged over 100 trials.

object tasks. For the challenging TransNAS-Bench-101-Micro Autoencoder and NAS-Bench-NLP PTB (Kombrink et al., 2011) tasks, ListMLE outperforms other ranking losses.

When the training portion becomes large, pairwise ranking losses such as BPR exhibit a dominant advantage in four tasks across three search spaces. This is consistent with the results in previous works (Xu et al., 2021; Hwang et al., 2024) that the performance predictor can achieve a much higher $\tau$ with pairwise ranking loss than MSE loss.

To better observe the performance of ranking losses as the training portion grows, we plot the $\tau$ of four ranking losses as representatives for different types of losses in Figure 3. We find that the pairwise and listwise ranking losses generalize better than the weighted and pointwise ranking ones because they optimize the overall ranking of architecture.

### 4.1.2 Performance in Top-Centered Rank Metrics

To evaluate the performance in recognizing top-performing architectures, we compare $\tau_w$, N@K and Rel@K of different ranking losses under 16 data splits in Figure 4. The K is set to 10 because we focus more on the top-performing architectures.

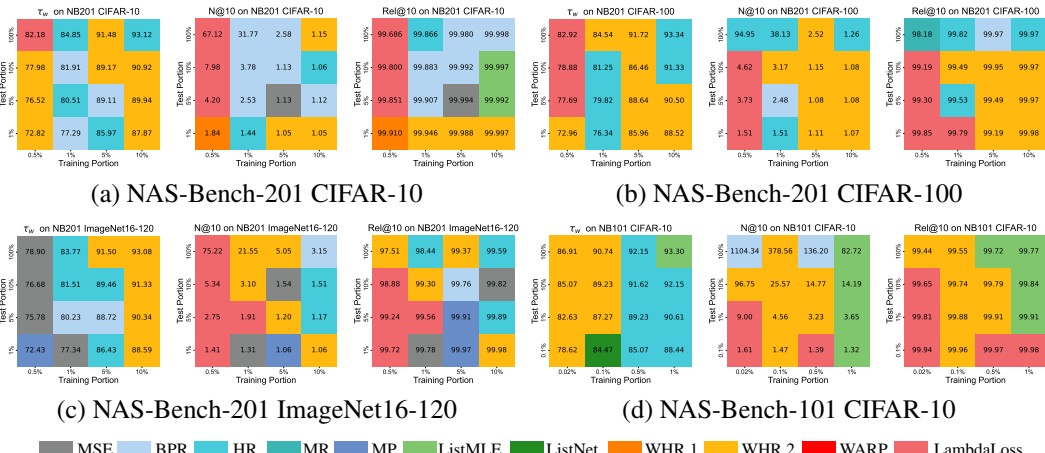

(a) NAS-Bench-201 CIFAR-10      (b) NAS-Bench-201 CIFAR-100

(c) NAS-Bench-201 ImageNet16-120      (d) NAS-Bench-101 CIFAR-10

Figure 4: $\tau_w$, N@10 and Rel@10 of different ranking loss functions on NAS-Bench-101 and NAS-Bench-201. Higher is better for $\tau_w$ and Rel@10 while lower is better for N@10. The results are averaged over 100 trials.

**Weighted Kendall Tau Results.** Figure 4 illustrates that WHR 2 achieves the highest $\tau_w$ under half of 16 data splits on NAS-Bench-201 CIFAR-10. Besides, BPR and HR maintain their competitive performance in $\tau$ and attain promising $\tau_w$ values. In addition, WHR 2 and LamadaLoss perform very well in $\tau_w$ but neither of them becomes the optimal loss for $\tau$ on the same task in Figure 2.

**N@K and Rel@K Results.** From Figure 4, we can see that weighted ranking loss shows excellent ability in N@10 and Rel@10. On NAS-Bench-201 CIFAR-10, WHR 2 achieves a very low N@10 of 1.15 with a training portion of 10% and a test portion of 100%. As the training portion grows, two pairwise ranking loss functions: BPR and HR yield impressive results in both metrics.

Figure 4 shows a very different trend from that of the overall ranking performance in Figure 2. Most Weighted ranking losses perform poorly in $\tau$ but exhibit high top-centered metrics especially when the training portion is low. Our results suggest that ranking losses which assign more weights to well-performing architectures can compensate for the limited training sample to identify top-performing architectures. Additionally, pairwise ranking losses such as BPR and HR can still achieve excellent results on top-centered metrics when the training portion gets larger. This indicates that pairwise ranking can leverage the ranking relationship between massive training samples to precisely recognize the top-performing architectures.

### 4.1.3 Enhancing Existing Predictors

To further investigate the power of ranking losses, we combine them with two representative performance predictors: NP (Wen et al., 2020) and PINAT (Lu et al., 2023). Both predictors have elaborately designed models but adopt the simple MSE loss function. In this part, BPR, ListMLE and WHR 2 loss are chosen as the representatives for their excellent performance in the rank-based metrics. The predictors are evaluated with four rank-based metrics for six data splits on NAS-Bench-101 and NAS-Bench-201. Note that the test portion is set to 100% in the following experiments. The results are averaged over 10 runs.

**Results on Kendall Tau.** Table 1 shows that changing the ranking loss in predictors can bring significant performance improvements in $\tau$. For example, PINAT and NP obtain a 6.62% and 12.04% gain with ListMLE loss on NAS-Bench-201 CIFAR-10 using a training portion of 0.5%, respectively. We also find that ListMLE loss helps NP realize a high $\tau$ of 0.68 with only 1% training samples, which even outperforms the result of MSE loss at a training portion of 10%.

| Datasets | NAS-Bench-101 CIFAR-10 | | | NAS-Bench-201 CIFAR-10 | | |
|---|---|---|---|---|---|---|
| Training portion | 0.02% | 0.1% | 1% | 0.5% | 1% | 10% |
| PINAT + MSE | 63.61±1.63 | 75.67±1.42 | 84.40±0.33 | 55.51±2.20 | 62.34±2.45 | 77.96±0.64 |
| PINAT + BPR | 66.21±0.71 | 76.75±1.04 | 84.66±0.02 | 61.32±2.09 | 67.11±1.81 | 85.18±0.07 |
| PINAT + ListMLE | **66.35±0.91** | **77.56±0.48** | **84.69±0.12** | **62.13 ±2.10** | **67.65±1.70** | **85.60±0.30** |
| PINAT + WHR 2 | 65.59±0.98 | 70.65±0.61 | 68.14±0.06 | 61.79 ±1.90 | 65.10±2.05 | 75.29±1.99 |
| NP + MSE | 52.99±1.63 | 63.14±0.05 | 72.71±0.02 | 52.14±10.69 | 52.20±5.83 | 66.26±1.30 |
| NP + BPR | 61.21±0.04 | 69.59±0.03 | 75.91±0.01 | 63.87±0.18 | 66.03±1.40 | 76.33±0.05 |
| NP + ListMLE | **62.06±0.33** | **70.41±0.22** | **78.92±0.07** | **64.18±2.32** | **68.22±1.64** | **80.26±0.35** |
| NP + WHR 2 | 48.40±0.91 | 53.49±0.97 | 60.53±0.56 | 62.45±1.24 | 63.91±1.05 | 73.67±0.72 |

Table 1: Kendall Tau (scaled up by a factor of 100, mean and standard deviation) on NAS-Bench-101 and NAS-Bench-201.

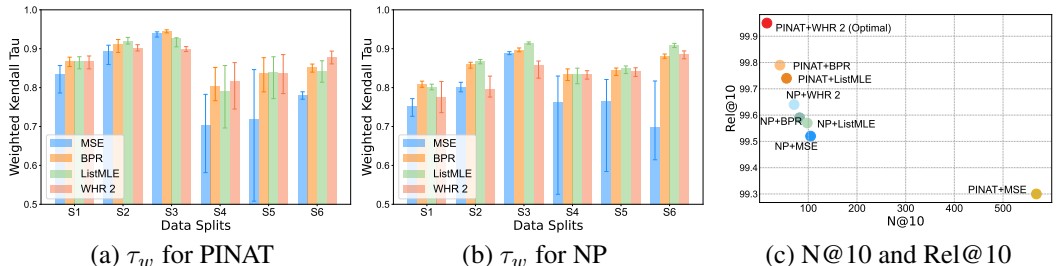

(a) $\tau_w$ for PINAT        (b) $\tau_w$ for NP        (c) N@10 and Rel@10

Figure 5: $\tau_w$, N@10 and Rel@10 for different ranking loss with PINAT and NP. S1 to S6 correspond to the six data splits in Table 1. The N@10 and Rel@10 are computed with a 1% training portion on NAS-Bench-201 CIFAR-10.

**Results on Top-centered Metrics.** From Figure 5(a) and Figure 5(b), we can see that both PINAT and NP roughly attain a higher $\tau_w$ with BPR and ListMLE loss. Meanwhile, NP encounters a severe performance drop with MSE loss on NAS-Bench-201 as the training portion grows from 1% to 10% (S5 to S6) because MSE loss is weak at leveraging massive training data in exploration of well-performing architectures. As for N@10 and Rel@10, the results in Figure 5(c) demonstrate the superior performance of WHR 2 loss that PINAT can almost discover the optimal architecture within top10 prediction at a budget of only 1% training data on NAS-Bench-201.

These results suggest that a suitable ranking loss can significantly enhance predictors' capability in correct overall architecture ranking and identifying top-performing architectures. The importance of the loss function can match the model design and the number of training samples for performance predictors.

### 4.2 EVALUATING PREDICTOR-BASED NAS

Now we investigate to what extent ranking loss functions facilitate predictor-based NAS methods. In this section, we select MSE, BPR, ListMLE and WHR 2 to represent the pointwise, pairwise, listwise and weighted ranking losses for their good performance in rank-based metrics, respectively. From previous experiments, we can see that in the low training portion region, pairwise and listwise ranking loss achieve higher $\tau$ while weighted ranking loss performs better in top-centered metrics. We opt for two popular predictor-based NAS frameworks: predictor-guided random search (Bergstra & Bengio, 2012) and predictor-guided evolutionary search (Real et al., 2019). All the results are averaged over 20 trials. Additional experimental results for predictor-based NAS frameworks can be found in Appendix C.2.

#### 4.2.1 PREDICTOR-BASED NAS FRAMEWORKS

We give a brief introduction to both frameworks and the detailed procedures are in Appendix A.2.

**Predictor-Guided Random Search.** This framework randomly samples candidate architectures from the search space several times and employs the predictor to select the top-ranked ones from

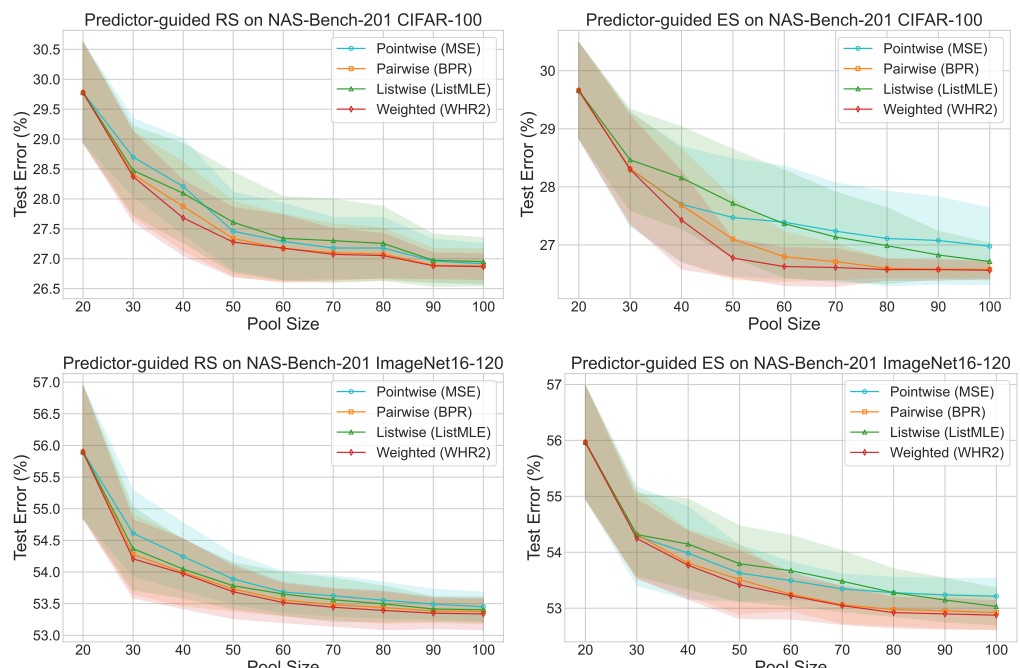

Figure 6: Test error vs. Pool size for the predictor-guided random search (RS) and the predictor-guided evolutionary search (ES) framework with different ranking losses on NAS-Bench-201.

| | BANANAS | CATE | DCLP | WeakNAS | Ours (ES) | | | |
|---|---|---|---|---|---|---|---|---|
| | | | | | MSE | BPR | ListMLE | WHR2 |
| # Queries | 150 | 150 | 300 | 200 | 150 | 150 | 150 | 150 |
| Test Error (%) | 5.92 | 5.88 | 5.83 | 5.82 | 5.95 | 5.86 | 5.84 | **5.81** |

Table 2: Comparison with previous works on NAS-Bench-101.

the candidates. Note that the predictor is trained on an initial architecture pool $P$ in advance. The newly chosen architectures are then added to $P$ and the predictor is updated upon the expanded architecture pool. This update process is iterated until the size of $P$ reaches the query budget. The best architecture in $P$ is chosen as the final search result.

**Predictor-Guided Evolutionary Search.** The predictor-guided evolutionary search framework also searches for promising architectures with an updated performance predictor. To expand the architecture pool during the search, we evolve the architectures in the pool to generate new candidate individuals and utilize the predictor to pick out ones that are predicted to perform well.

### 4.2.2 RESULTS

**Results on NAS-Bench-201.** The results on NAS-Bench-201 are presented in Figure 6. We find that weighted ranking loss achieves the lowest test error under both frameworks. For instance, WHR 2 outperforms other ranking losses consistently in CIFAR-100 using the predictor-guided ES framework. Although BPR loss also yields favorable results with the full query budget, WHR 2 loss has a significant advantage in the region of low pool size.

**Results on NAS-Bench-101.** We adopt a predictor-guided evolutionary search framework and compare our method with previous SotA predictors: BANANAS (White et al., 2021), CATE (Yan et al., 2021), WeakNAS (Wu et al., 2021) and DCLP (Zheng et al., 2024). From Table 2, We find a major decrease in test error when changing the loss function from MSE to WHR 2. Meanwhile, WHR 2 loss helps our method outperform the previous SotA methods with the fewest queries. Note that the predictor used in our method is only composed of a basic GCN encoder and a simple MLP regressor.

| Tasks | Cls.O. | Cls.S. | Auto. | Jigsaw | Avg. Rank |
|---|---|---|---|---|---|
| Metric | Acc.$^\uparrow$ | Acc.$^\uparrow$ | SSIM$^\uparrow$ | Acc.$^\uparrow$ | |
| RS (Bergstra & Bengio, 2012) | 45.16 | 54.41 | 55.94 | 94.47 | 61.00 |
| REA (Real et al., 2019) | 45.39 | 54.62 | **56.96** | 94.62 | 27.50 |
| MSE | 45.36 | 54.66 | 55.59 | 94.75 | 41.25 |
| BPR | 45.47 | 54.76 | 56.20 | 94.77 | 24.25 |
| ListMLE | 45.53 | 54.66 | 55.95 | 94.63 | 35.5 |
| WHR 2 | **45.66** | **54.83** | 56.57 | **94.84** | **15.75** |
| Global Best | 46.32 | 54.94 | 57.72 | 95.37 | 1 |

Table 3: Searching results on TransNAS-Bench-101-Micro.

| Predictor | Test Err.(%) | Params(M) | GPU Days | Search Method | Loss Function |
|---|---|---|---|---|---|
| TNASP (Lu et al., 2021) | 2.57±0.04 | 3.6 | 0.3 | Evolution | MSE |
| CDP (Liu et al., 2022) | 2.63±0.08 | 3.3 | 0.1 | Random | MSE |
| NPENAS (Wei et al., 2022) | 2.54±0.10 | 3.5 | 1.8 | Evolution | MSE |
| PINAT (Lu et al., 2023) | 2.54±0.08 | 3.6 | 0.3 | Evolution | MSE |
| | 2.74±0.04 | 4.4 | 0.14 | Random | MSE |
| Ours (GCN + MLP) | 2.57±0.04 | 3.6 | 0.14 | Random | BPR |
| | 2.62±0.07 | 3.8 | 0.14 | Random | ListMLE |
| | **2.48±0.04** | 4.3 | 0.14 | Random | WHR 2 |

Table 4: Performance comparison between the architectures searched by different ranking loss functions and other predictor-based methods on CIFAR-10.

**Results on TransNAS-Bench-101-Micro.** We also search for architectures on TransNAS-Bench-101-Micro using a predictor-guided evolutionary framework and compare with baseline methods in Table 3. The query budget is set to 50. Among all, WHR 2 loss attains the best average rank on four tasks. Meanwhile, we find the average rank of MSE and ListMLE losses are inferior to the baselines. This performance gap indicates that weighted ranking loss can significantly boost the accuracy of the predictor and lead to a promising solution in the search space.

**Results on DARTS.** Table 4 demonstrates the results of the architectures discovered by four representative ranking losses on DARTS search space for CIFAR-10. We find that WHR 2 loss achieves the lowest test error rate of 2.48% in only 0.14 GPU Days, which outperforms previous SotA predictor-based methods. Besides, pairwise and listwise ranking loss perform better than MSE loss within the same predictor, which reflects the significance of the ranking loss function for predictor-based NAS methods. This suggests that when the training samples for performance predictors are limited, the weighted ranking loss is the optimal choice in pursuit of a promising architecture for new tasks because of their excellent performance on top-centered rank metrics.

## 5 CONCLUSION

In this paper, we provide the first comprehensive study for 11 ranking loss functions in performance predictors, including pointwise, pairwise, listwise and weighted ranking loss. We compare their performance in ranking correlation metrics and top-centered metrics under various settings. Meanwhile, we combine these ranking loss functions into predictor-based NAS methods to search for promising architectures. We find ranking loss with excellent performance on top-centered rank metrics can help find architectures with high quality. In addition, our experiments show that a well-designed ranking loss can greatly improve the performance of existing predictors and predictor-based NAS methods. In the future, we will study some complex ranking loss functions which are composed of several single losses to further enhance performance predictors.

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

APPENDIX

The appendix includes detailed information on the materials that are not covered in the main paper for the page limit and additional experimental results. The organization of this file is presented as follows:

- Appendix A - We elaborate on the details of different ranking loss functions and predictor-based NAS frameworks used in our experiments.
- Appendix B - We provide detailed experiment settings including search space and hyper-parameter configuration.
- Appendix C - We provide additional experimental results for the evaluation of the performance predictors, predictor-based NAS methods, computational costs and visualization results.

## A    DETAILED DESCRIPTIONS OF BASELINE METHODS

### A.1    DETAILED DESCRIPTIONS OF RANKING LOSS FUNCTIONS

Now we give detailed descriptions of 11 different ranking loss functions that we used.

**Mean Square Error (MSE).** MSE loss is the most widely used loss function in existing performance predictors (Wen et al., 2020; Lu et al., 2021; Liu et al., 2022; Lu et al., 2023). They first predict absolute architecture performance precisely and then rank architectures according to the predicted values. We directly use the official implementation from the Pytorch package.

**Bayesian Personalized Ranking (BPR).** BPR loss (Rendle et al., 2009) is very popular in the field of recommendation system and was first combined with the performance predictor by GMAE-NAS (Jing et al., 2022). For each sample in a batch, BPR compares its predicted score with other samples which have higher true accuracy. BPR optimizes the relative ranking between architecture pairs, encouraging the predictor to capture the difference between high-rank and low-rank architectures for better ranking ability. We use the implementation from the open source of GMAE-NAS.

**ListMLE.** ListMLE loss (Xia et al., 2008) is a classic listwise ranking loss in information retrieval and was introduced into performance predictor by DCLP (Zheng et al., 2024). ListMLE loss aims to maximize the likelihood of the correct architecture ranking, making the predictor better estimate the ranking order more accurately. We use the implementation from the open source of DCLP.

**Margin Rank (MR).** MR loss was used in FlowerFormer (Hwang et al., 2024) to optimize the ranking of architecture pairs. It promotes correct ranking by comparing pairs of architecture scores and maintaining a minimum margin between the high-rank architecture and the low-rank one. We directly use the official implementation from the Pytorch package.

**Hinge Rank (HR).** HR loss can be categorized into pairwise ranking and was applied to performance predictor by ReNAS (Xu et al., 2021) to predict relativistic architecture performance. A hinge function is used to calculate the loss. The loss will be 0 only when the architecture pair is correctly ranked and the score difference is within the margin. We use the implementation from the open source of ReNAS.

**Weighted Hinge Rank (WHR1 & WHR2).** We design two versions of weighted hinge ranking loss functions based on Hinge Rank by assigning more weights to architectures with better performance. Given a pair of architecture scores $s_i$ and $s_j$, the loss function is calculated by $\mathcal{L}_{\mathcal{HR}} \times w_i \times w_j$, where $w_i$ and $w_j$ are the weights for two architectures. For WHR1, the weight of each architecture $X$ is set to $(N - rank(X) + 1)/N$, where $rank(X)$ denotes the true ranking of X. As for WHR2, the weight of each architecture $X$ is formulated as $exp(s_X) - 1$. WHR2 assigns a higher weight to top-performing architectures compared to WHR1. We implement two WHR loss functions on the foundation of Hinge Rank.

**ListNet.** ListNet loss (Xia et al., 2008) is another classic listwise ranking loss that converts both the predicted scores and true scores into probability distributions with the Softmax function and calculates the cross-entropy between them to enhance the model's ranking ability. We develop the original code of ListNet to fit the task of architecture performance.

**Weighted Approximate-Rank Pairwise (WARP).** WARP loss (Weston et al., 2011) was originally designed for image annotation. To calculate the loss, pairs of items are randomly sampled until a wrongly ranked pair is found and the predicted scores of the two incorrect items. A weight will be assigned to the loss according to the times it takes to find a wrongly ranked pair. Based on this idea, we tailor the WARP loss for the performance predictor. For each architecture in the batch, we randomly choose others to construct pairs until a pair is ranked incorrectly. If the incorrect pair appears soon, we think the predictor is poor at ranking and will assign a large weight to this sample when calculating the loss. Instead, if we need more samples to find a wrong pair, the performance of the predictor is already good and the loss should be small. We borrow the core idea of the original WARP and implement a new version for architecture performance ranking.

**Magnitude Preserving (MP).** MP loss (Cortes et al., 2007) combines the idea of MSE loss and pairwise ranking loss and preserves the magnitude of ground truth. MP loss aims to make the difference between the predicted scores approach that between the true accuracy within each data pair for better ranking ability. So the correctly ranked pairs will still be penalized. We develop the original code of MP and apply it to predict architecture performance.

**LambdaLoss.** LambdaLoss (Wang et al., 2018) is a probabilistic framework for ranking metric optimization advanced from the classic LambdaRank (Burges et al., 2006), which contains different metric-driven loss functions for the model. To enhance the model's ability to identify top-performing architectures, we design the loss function based on the one that was originally used to optimize the model performance in Normalized Discounted Cumulative Gain (NDCG), a top-rank centered metric. We use the implementation from the AllRank framework (Pobrotyn et al., 2020).

### A.2 ALGORITHMS OF PREDICTOR-BASED NAS FRAMEWORKS

Now we provide detailed algorithms of the two predictor-based NAS frameworks that we employed in Algorithm 1 and Algorithm 2.

---

**Algorithm 1:** Predictor-guided Random Search Framework

---

**Initialize:** Sample a few architectures randomly from search space $S$ to construct an initialization space $S_0$, train them for ground-truth performance and add them to $history$.
**while** $|history| + |S_0| < N$ **do**
  Train predictor using architectures in $history$ ;
  Sample $M$ candidate architectures at random from $S - history$ ;
  Utilize predictor to predict the performance of each candidate architecture ;
  Update $history$ by adding architectures with top-$K$ predicted values ;
**return** Architecture $A'$ with best ground-truth performance in $history$

---

**Algorithm 2:** Predictor-guided Evolutionary Search Framework

---

**Initialize :** Sample a few architectures randomly from search space $S$ to construct an initialization space $S_0$, train them for ground-truth performance and add them to $history$ and population $P$.
**while** $|history| + |S_0| < N$ **do**
  Train the predictor using architectures in $history$ ;
  Select T well-performing architectures in $P$ through Binary Tournament Selection ;
  Mutate the selected architectures and generate M candidate architectures ;
  Utilize predictor to predict the performance of each candidate architecture ;
  Update $history$ and $P$ by adding architectures with top-$K$ predicted values ;
  Remove the oldest architecture from $P$;
**return** Architecture $A'$ with best ground-truth performance in $history$

---

# B DETAILED EXPERIMENTAL SETTINGS

## B.1 SEARCH SPACES

We discuss the search spaces we used in the following part.

**NAS-Bench-101.** NAS-Bench-101 (Ying et al., 2019) is a cell-based search space composed of over 423k architectures. An operation is represented by a node in the cell. Each architecture cell includes at most seven nodes and nine edges. The search space provides the accuracy of architectures on the CIFAR-10 dataset.

**NAS-Bench-201.** NAS-Bench-201 (Dong & Yang, 2019) is also a cell-based search space which contains 15625 architectures. The operation type is represented by the edge on NAS-Bench-201. There exist four nodes and six edges in the architecture cell. NAS-Bench-201 provides the accuracy of architectures on the CIFAR-10, CIFAR-100 and ImageNet16-120 datasets.

**TransNAS-Bench-101-Micro & Macro.** TransNAS-Bench-101 (Duan et al., 2021) is composed of two parts: a micro (cell-based) search space containing 4096 architectures and a macro (skeleton-based) search space containing 3256 architectures. The constitution of the architecture cell in the micro search space is the same as NAS-Bench-201. As for the macro space, we use the same encoding in Arch-Graph (Huang et al., 2022). The search space provides architecture performance across diverse tasks including segmentation, regression, pixel-level prediction and self-supervised tasks. In our experiments, we test our methods on four tasks: Class-Object, Class-Scene, Jigsaw and Autoencoder.

**NAS-Bench-NLP.** NAS-Bench-NLP contains 14332 architectures and a cell-based search space. There are at most 24 nodes and 26 edges in each cell. Different from the other search spaces, this one aims to design well-preforming architectures for NLP tasks. NAS-Bench-NLP provides test results of the architectures which are trained on Penn Tree Bank dataset (Kombrink et al., 2011) for 50 epochs.

**DARTS.** DARTS (Liu et al., 2018b) is an open-domain cell-based search space without available architecture performance. It is the largest search space in our experiments which contains around $10^{18}$ architectures. Each architecture consists of a normal cell and a reduction cell, which contains seven nodes and eight edges in a single cell.

| Search Space | Max Node / Edge Num. | Size | Is Cell-based? | URL |
|---|---|---|---|---|
| NAS-Bench-101 | 7 / 9 | 423k | ✓ | https://github.com/google-research/nasbench |
| NAS-Bench-201 | 4 / 6 | 15625 | ✓ | https://github.com/D-X-Y/NAS-Bench-201 |
| TransBench-101-Micro | 4 / 6 | 4096 | ✓ | https://github.com/yawen-d/TransNASBench |
| TransBench-101-Macro | 8 / 7 | 3256 | ✗ | |
| NAS-Bench-NLP | 24 / 26 | 14322 | ✓ | https://github.com/fmsnew/nas-bench-nlp-release |
| DARTS | 7 / 14 | $10^{18}$ | ✓ | https://github.com/quark0/darts |

Table 5: Basic information of different search spaces used in our experiments.

The URLs of search spaces are reported in Table 5.

## B.2 HYPERPARAMETER SETTINGS

| Search Space | Query Budget (N) | Candidate Number (M) | Update Number (K) | Initial Size ($|S_0|$) |
|---|---|---|---|---|
| NAS-Bench-101 | 150 | 200 | 10 | 20 |
| NAS-Bench-201 | 100 | 200 | 10 | 20 |
| TransBench-101-Micro | 50 | 100 | 5 | 20 |

Table 6: Hyperparameter settings for predictor-based NAS methods in our experiments.

For all the ranking and searching experiments, we combine different ranking loss functions with a simple GCN encoder to construct our predictor for a fair comparison. The predictor is composed of a four-layer GCN and a three-layer MLP. The hyperparameter setting of the predictor is the same across different search spaces and tasks. The batch size is set to one-tenth of the training number.

| | NB101-CF10 | TB101_MICRO-AUTO | TB101_MACRO-AUTO | TB101_MICRO-OBJECT | TB101-MACRO_OBJECT | TB101_MICRO-SCENE | TB101_MACRO-SCENE | TB101_MICRO-JIGSAW | TB101_MACRO_JIGSAW | NBNLP-PTB | NB201-CF10 | NB201-CF100 | NB201-IMGNT |
|---|---|---|---|---|---|---|---|---|---|---|---|---|---|
| MSE | 85.86 | 13.47 | -9.23 | 44.59 | 65.35 | 53.82 | 72.01 | 41.09 | 64.04 | 23.22 | 78.56 | 80.98 | 78.90 |
| BPR | 85.80 | 62.88 | 83.59 | 45.90 | 67.88 | 57.14 | 72.90 | 43.16 | 73.08 | 28.35 | 80.55 | 80.25 | 77.13 |
| HR | 85.77 | 62.96 | 83.60 | 46.35 | 67.50 | 56.94 | 72.66 | 43.44 | 72.73 | 28.04 | 80.57 | 80.48 | 77.27 |
| MR | 85.98 | 63.00 | 83.57 | 46.68 | 67.91 | 57.44 | 73.20 | 43.42 | 72.89 | 27.70 | 80.97 | 80.62 | 77.58 |
| MP | 84.75 | 31.95 | 67.65 | 44.49 | 64.65 | 53.91 | 70.77 | 40.47 | 62.12 | 25.11 | 75.22 | 79.04 | 78.63 |
| ListMLE | 83.74 | 62.82 | 83.53 | 46.19 | 66.10 | 56.73 | 71.99 | 43.92 | 71.50 | 28.79 | 79.70 | 79.58 | 76.70 |
| ListNet | 85.38 | -8.22 | 49.58 | 43.62 | 66.99 | 54.70 | 72.85 | 39.06 | 68.28 | 24.17 | 80.10 | 80.46 | 76.98 |
| WHR 1 | 85.90 | 62.43 | 83.54 | 45.21 | 68.05 | 57.41 | 73.26 | 42.96 | 74.35 | 22.32 | 80.66 | 80.56 | 76.02 |
| WHR 2 | 86.91 | -4.35 | 4.29 | 40.79 | 64.42 | 57.30 | 70.31 | 38.97 | 75.01 | 16.27 | 82.08 | 82.16 | 77.97 |
| WARP | 84.17 | 37.38 | 70.60 | 46.82 | 66.90 | 59.34 | 67.64 | 36.76 | 68.17 | 22.58 | 80.81 | 81.17 | 76.63 |
| LambdaLoss | 86.85 | 56.87 | 81.17 | 44.66 | 67.10 | 60.23 | 67.27 | 34.29 | 68.03 | 23.24 | 82.15 | 82.92 | 77.63 |

Figure 7: Weighted Kendall Tau of different ranking loss functions in 13 different tasks across four search spaces (scaled up by a factor of 100).

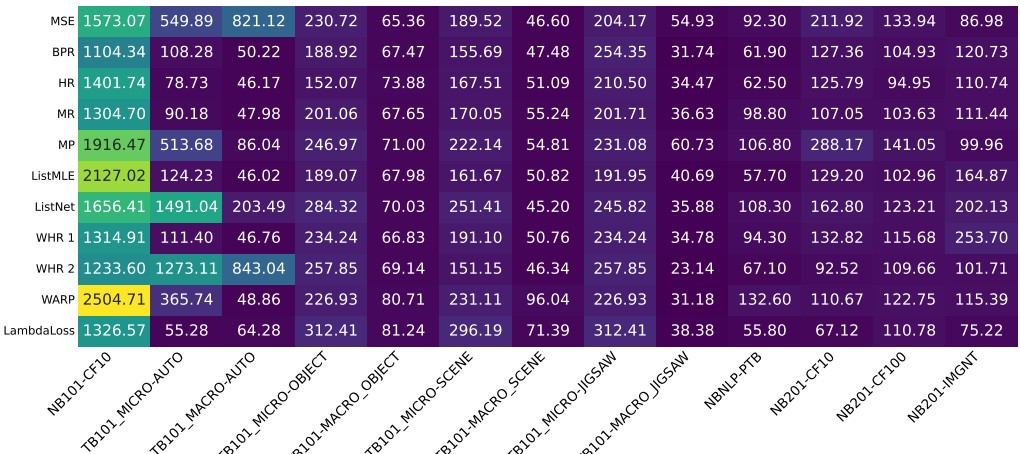

| | NB101-CF10 | TB101_MICRO-AUTO | TB101_MACRO-AUTO | TB101_MICRO-OBJECT | TB101-MACRO_OBJECT | TB101_MICRO-SCENE | TB101_MACRO-SCENE | TB101_MICRO-JIGSAW | TB101_MACRO_JIGSAW | NBNLP-PTB | NB201-CF10 | NB201-CF100 | NB201-IMGNT |
|---|---|---|---|---|---|---|---|---|---|---|---|---|---|
| MSE | 1573.07 | 549.89 | 821.12 | 230.72 | 65.36 | 189.52 | 46.60 | 204.17 | 54.93 | 92.30 | 211.92 | 133.94 | 86.98 |
| BPR | 1104.34 | 108.28 | 50.22 | 188.92 | 67.47 | 155.69 | 47.48 | 254.35 | 31.74 | 61.90 | 127.36 | 104.93 | 120.73 |
| HR | 1401.74 | 78.73 | 46.17 | 152.07 | 73.88 | 167.51 | 51.09 | 210.50 | 34.47 | 62.50 | 125.79 | 94.95 | 110.74 |
| MR | 1304.70 | 90.18 | 47.98 | 201.06 | 67.65 | 170.05 | 55.24 | 201.71 | 36.63 | 98.80 | 107.05 | 103.63 | 111.44 |
| MP | 1916.47 | 513.68 | 86.04 | 246.97 | 71.00 | 222.14 | 54.81 | 231.08 | 60.73 | 106.80 | 288.17 | 141.05 | 99.96 |
| ListMLE | 2127.02 | 124.23 | 46.02 | 189.07 | 67.98 | 161.67 | 50.82 | 191.95 | 40.69 | 57.70 | 129.20 | 102.96 | 164.87 |
| ListNet | 1656.41 | 1491.04 | 203.49 | 284.32 | 70.03 | 251.41 | 45.20 | 245.82 | 35.88 | 108.30 | 162.80 | 123.21 | 202.13 |
| WHR 1 | 1314.91 | 111.40 | 46.76 | 234.24 | 66.83 | 191.10 | 50.76 | 234.24 | 34.78 | 94.30 | 132.82 | 115.68 | 253.70 |
| WHR 2 | 1233.60 | 1273.11 | 843.04 | 257.85 | 69.14 | 151.15 | 46.34 | 257.85 | 23.14 | 67.10 | 92.52 | 109.66 | 101.71 |
| WARP | 2504.71 | 365.74 | 48.86 | 226.93 | 80.71 | 231.11 | 96.04 | 226.93 | 31.18 | 132.60 | 110.67 | 122.75 | 115.39 |
| LambdaLoss | 1326.57 | 55.28 | 64.28 | 312.41 | 81.24 | 296.19 | 71.39 | 312.41 | 38.38 | 55.80 | 67.12 | 110.78 | 75.22 |

Figure 8: N@10 of different ranking loss functions in 13 different tasks across four search spaces.

For example, if the training number is 424 on NAS-Bench-101, the batch size should be 43. The predictor is trained with a learning rate of 1e-3, a weight decay of 1e-3 and a dropout of 0.15. The number of epochs is set to 300 for most ranking losses except for WARP and LambdaLoss. They use 30 epochs by default because a longer training epoch leads to very poor performance on all the rank-based metrics. The configuration for the search experiments is included in the Table 6. All the experiments are conducted on a single RTX 3090.

## C  ADDITIONAL EXPERIMENTS RESULTS

### C.1  FULL RESULTS FOR THE EVALUATION OF PERFORMANCE PREDICTORS

In Figure 7, Figure 8 and Figure 9, we plot the weighted Kendall Tau, N@10 and Rel@10 of different ranking loss functions in 13 tasks. The training portion is set to the smallest in the respective range of data splits and the test portion is set to 100%.

In Figure 10 and Figure 11, we provide the complete comparison between 11 ranking loss functions in 12 tasks. In Figure 12, we plot the performance of ranking loss functions in Kendall Tau and Weighted Kendall Tau as the training portion grows. The test portion is fixed to 100%. For the page limit, the results on NAS-Bench-NLP are included in Figure 13.

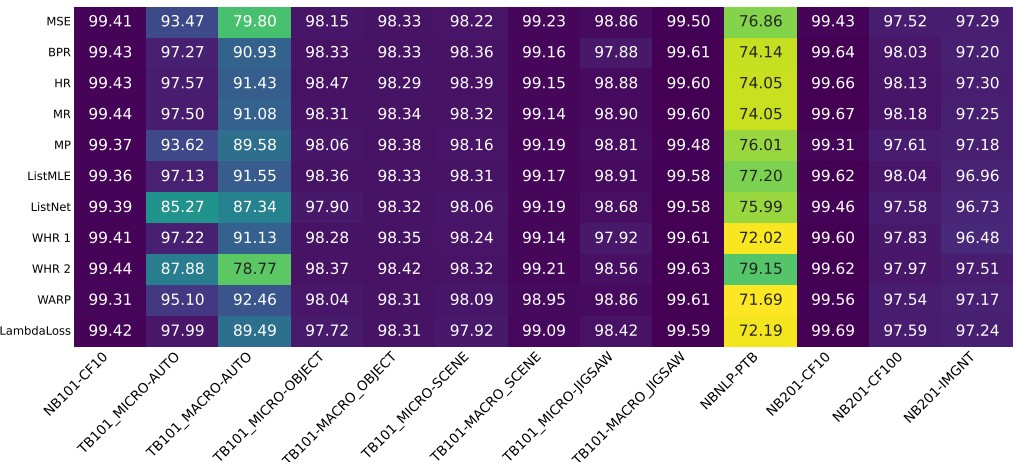

Figure 9: Rel@10 of different ranking loss functions in 13 different tasks across four search spaces (scaled up by a factor of 100).

## C.2 FULL RESULTS FOR THE EVALUATION OF PREDICTOR-BASED NAS METHODS

We plot the search results of predictor-based NAS frameworks using different ranking losses on NAS-Bench-201 CIFAR-10 and NAS-Bench-101 in Figure 14 and Figure 15.

## C.3 COMPUTATIONAL COST

| Loss Function | MSE | BPR | HR | MR | MP | ListMLE | ListNet | WHR 1 | WHR 2 | WARP | LambdaLoss |
|---|---|---|---|---|---|---|---|---|---|---|---|
| Training Time | 55.21 | 110.17 | 54.64 | 55.65 | 56.67 | 58.02 | 56.11 | 55.86 | 55.71 | 24.98 | 6.27 |
| Inference Time | 39.24 | 38.94 | 37.58 | 37.91 | 37.97 | 38.70 | 38.91 | 37.79 | 38.90 | 38.91 | 38.54 |

Table 7: Training and inference time on NAS-Bench-101 with a training portion of $0.1\%$ and a test portion of $100\%$.

| Loss Function | MSE | BPR | ListMLE | WHR 2 |
|---|---|---|---|---|
| Training Time | 750.6 | 991.5 | 745.8 | 751.3 |
| Inference Time | 253.2 | 254.7 | 250.5 | 252.8 |

| Loss Function | MSE | BPR | ListMLE | WHR 2 |
|---|---|---|---|---|
| Training Time | 110.5 | 190.6 | 117.3 | 111.5 |
| Inference Time | 42.1 | 41.7 | 40.8 | 42.2 |

Table 8: Training and inference time on NAS-Bench-101 with a training portion of $0.1\%$ and a test portion of $100\%$ for PINAT.

Table 9: Training and inference time on NAS-Bench-101 with a training portion of $0.1\%$ and a test portion of $100\%$ for NP.

We also calculate the computational cost of all the ranking loss functions on NAS-Bench-101 in Table 7. Among them, BPR is the most time-consuming option and takes two times longer than MSE when training. In contrast, WARP and LambdaLoss have very low training time because they require fewer epochs to train. In general, it is still very fast to finish training and inference because of the simple predictor.

We also calculate the computation cost for two representative predictors (Lu et al., 2023; Wen et al., 2020) in Table 8 and Table 9. The trend is similar to Table 7 but the time increases a lot for all losses. This indicates that the computational cost is more related to the composition of the predictor while the choice of loss function makes little difference.

## C.4 VISUALIZATION OF SEARCH RESULTS ON DARTS

We visualize the architecture cells searched by different ranking losses in Figure 16.

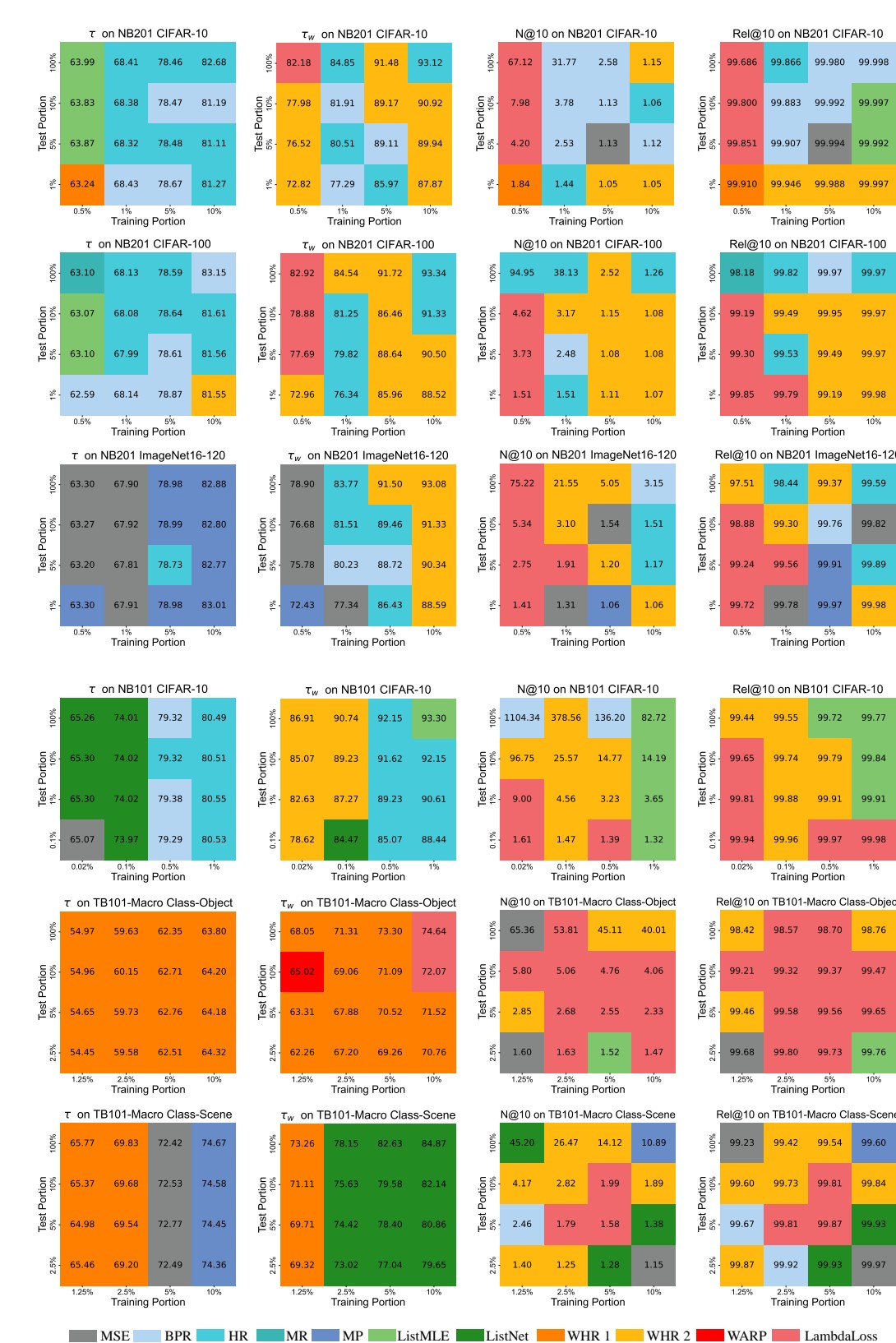

Figure 10: Kendall Tau of different ranking loss functions under various settings ($\tau$, $\tau_w$ and Rel@10 are scaled up by a factor of 100). The results are averaged over 100 trials.

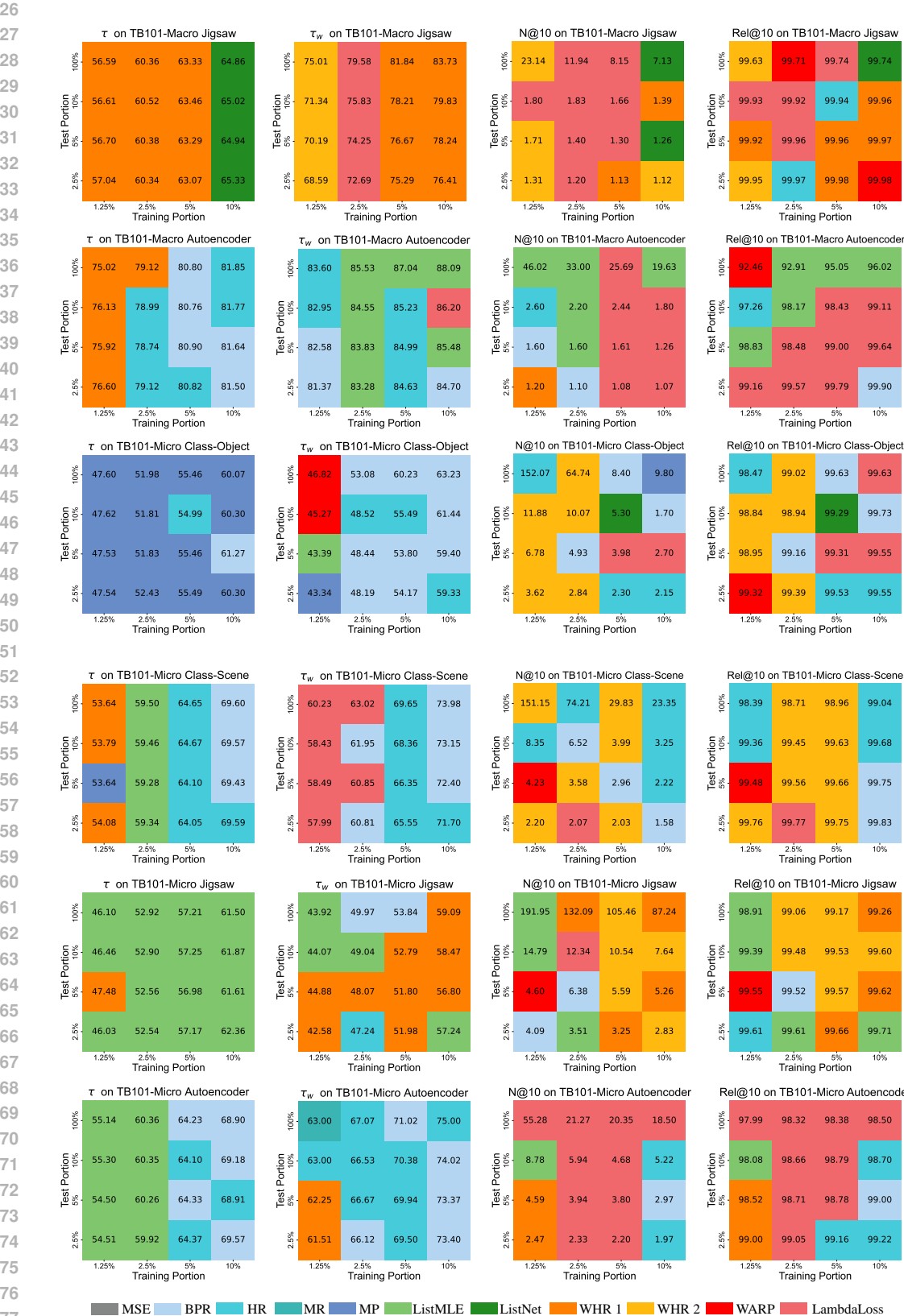

Figure 11: Results of different ranking loss functions under various settings ($\tau$, $\tau_w$ and Rel@10 are scaled up by a factor of 100). The results are averaged over 100 trials.

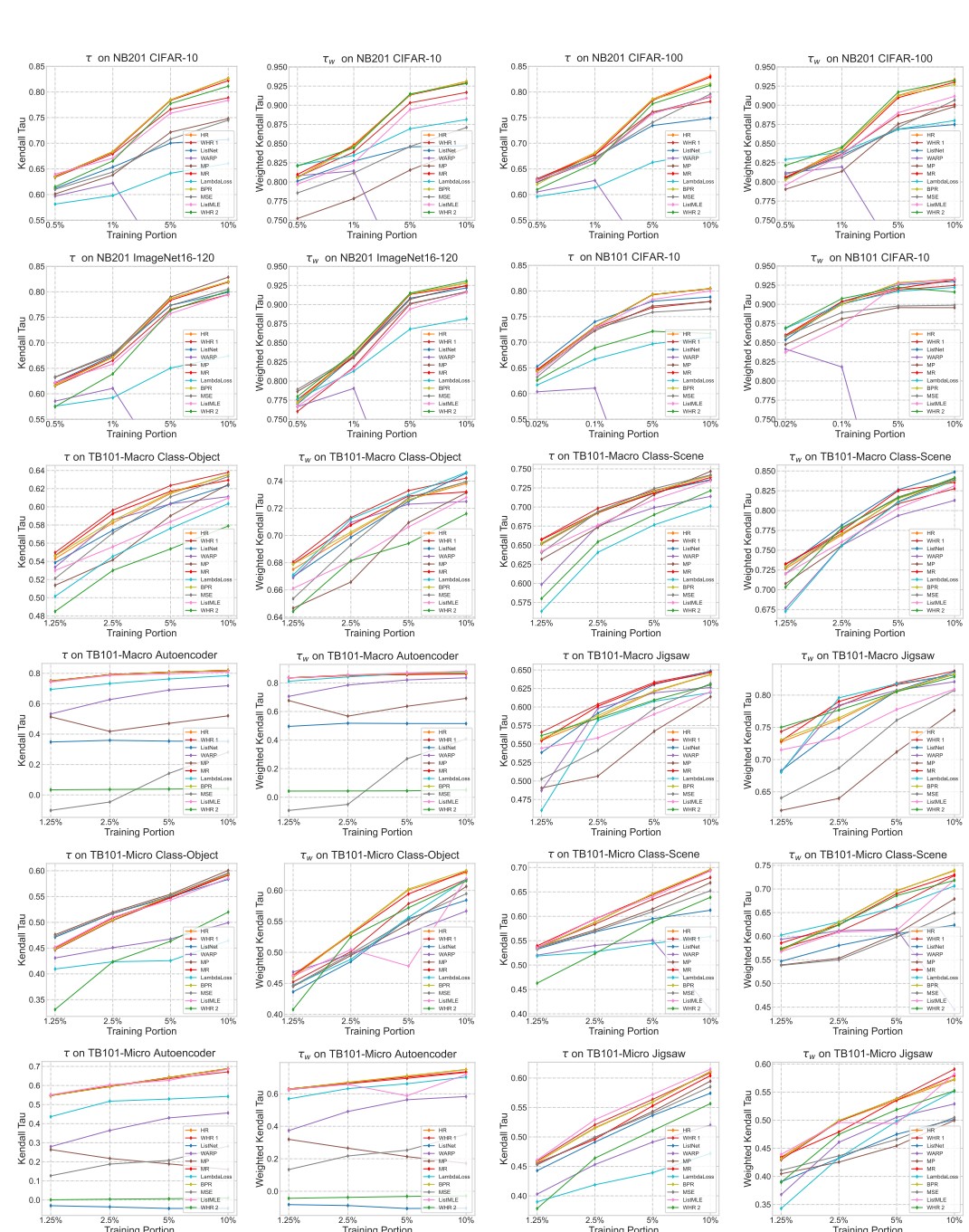

Figure 12: $\tau$ and $\tau_w$ of 11 ranking loss functions as the training portion grows. The test portion is fixed to $100\%$. The results are averaged over 100 trials.

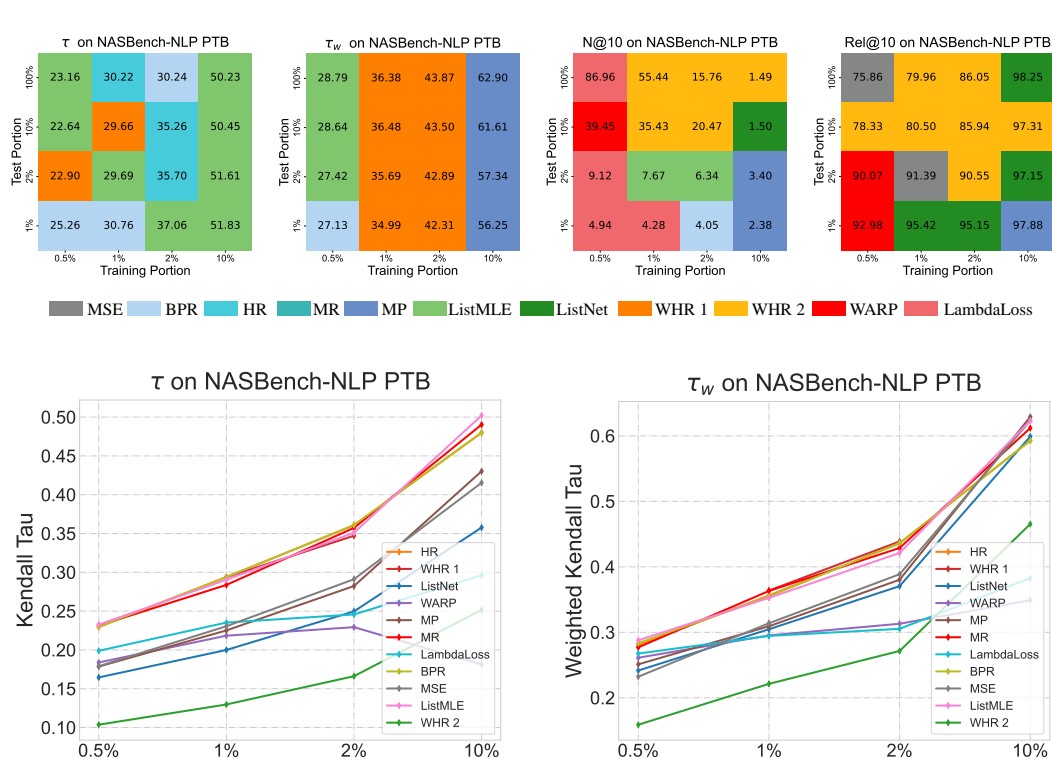

Figure 13: Results of different ranking loss functions on NAS-Bench-NLP. The results are averaged over 100 trials.

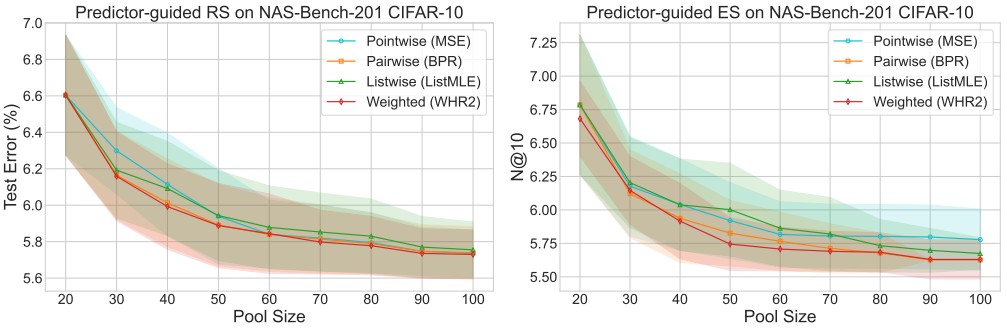

Figure 14: Test error vs. Pool size for the predictor-guided random search (RS) and the predictor-guided evolutionary search (ES) framework with different ranking losses on NAS-Bench-201 CIFAR-10.

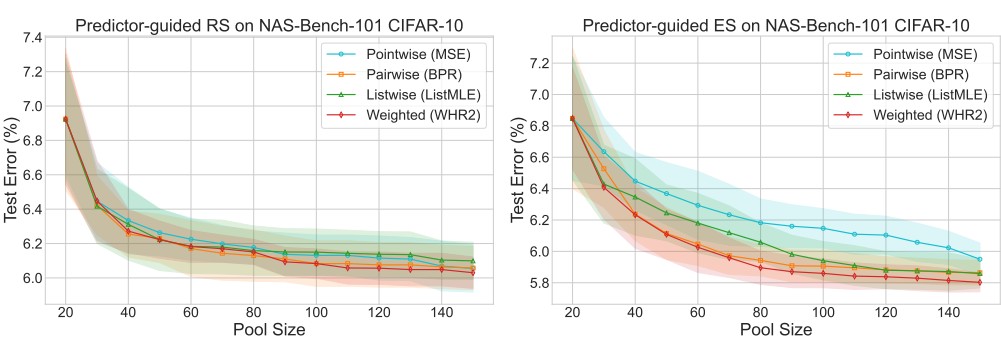

Figure 15: Test error vs. Pool size for the predictor-guided random search (RS) and the predictor-guided evolutionary search (ES) framework with different ranking losses on NAS-Bench-101.

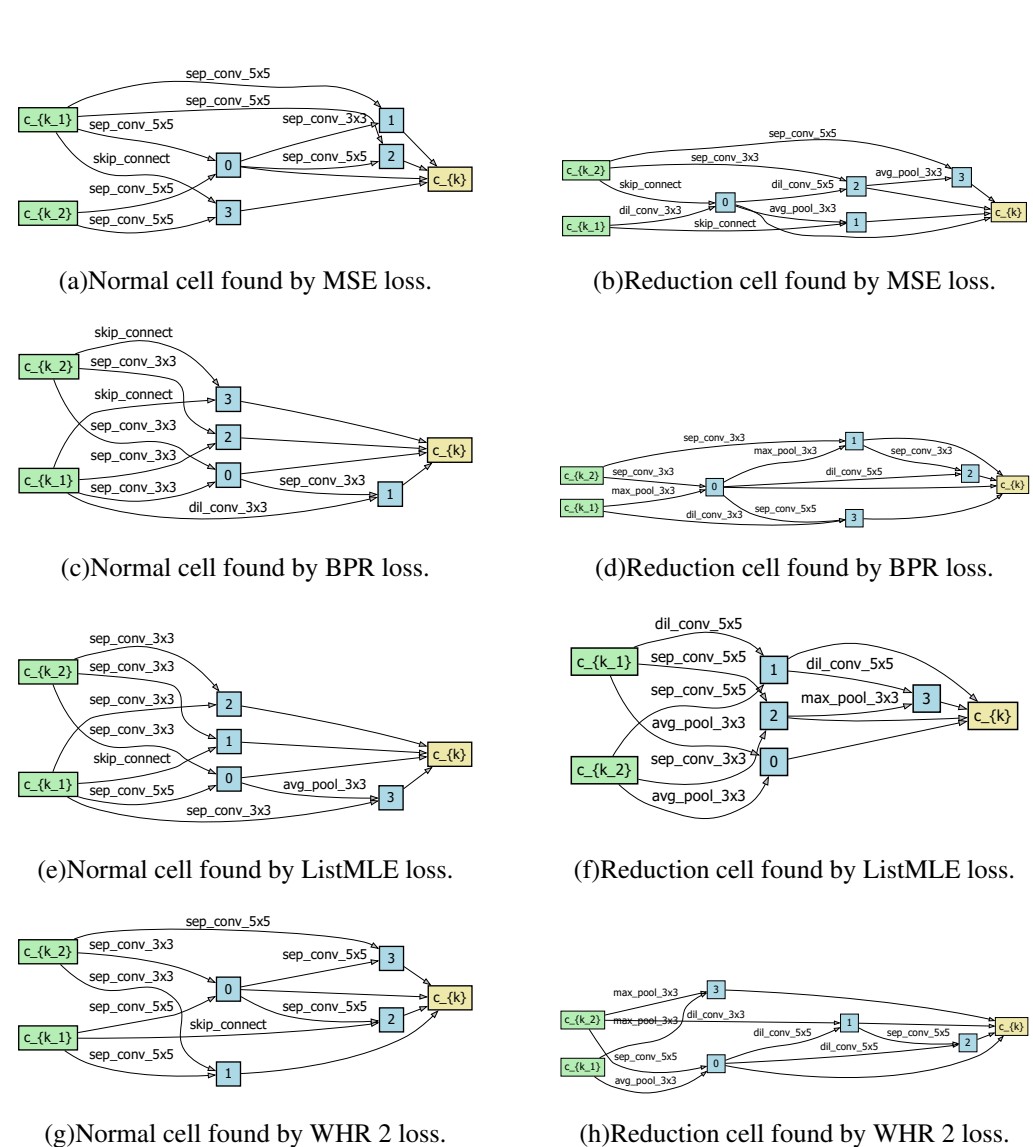

Figure 16: Cell architectures found by different ranking losses on CIFAR-10 dataset.