# OpenReview forum: "Evaluating Ranking Loss Functions in Performance Predictor for NAS"
_ICLR.cc/2025/Conference — ICLR 2025 Conference Withdrawn Submission_

### Official Review · Reviewer_mcru · 2024-10-22

**Soundness:** 3
**Presentation:** 2
**Contribution:** 3
**Rating:** 5
**Confidence:** 4

**Summary:**

This paper aims to provide a comprehensive benchmark and detailed analysis of ranking loss functions for training performance predictors in neural architecture search (NAS). Specifically, the authors compare 11 ranking loss functions (including pointwise, pairwise, listwise, and weighted ranking loss) across 5 NAS search spaces and 13 corresponding NAS tasks. Notably, the authors employ various evaluation metrics, including global, rank-weighted, and Top-$K$ metrics, emphasizing the importance of using top-centered metrics for NAS tasks. Additionally, the authors evaluate the practical performance of performance predictors trained with each loss function on two NAS frameworks.

**Strengths:**

- The paper systematically studies the application of ranking loss functions in NAS, which is an important and noteworthy issue.
- The paper conducts fair performance comparisons among 11 ranking loss functions, including pointwise, pairwise, listwise, and weighted ranking loss, covering most types of ranking loss functions.
- The paper employs various evaluation metrics, including the traditional global metric Kendall Tau, as well as ranking-based metrics like Weighted Kendall Tau, Top-$K$ metrics N@$K$, and Rel@$K$, providing a comprehensive assessment of the loss functions' performance.
- The paper conducts extensive experiments to benchmark the effectiveness of ranking loss functions on NAS tasks, accompanied by detailed analysis.
- The structure of the paper is clear and straightforward.

**Weaknesses:**

However, before this paper can be accepted, I still have the following **major concerns**:

**Presentation:** The definitions of the loss functions and evaluation metrics are very vague, which is detrimental to reproducibility. While some intuitive explanations are provided in Section 3, the lack of formal mathematical definitions in Appendix A is quite confusing.

- For example, in the definition of Weighted Approximate-Rank Pairwise (WARP) in Appendix A, the authors state, "If the incorrect pair appears soon, we think the predictor is poor at ranking and will assign a large weight to this sample when calculating the loss". How exactly is this weight calculated? I couldn't find the details anywhere in this paper.
- Another example is the even more ambiguous definition of metrics in Section 3.2. For instance, I can't understand the statement in Weighted Kendall Tau about "There is a hyperbolic drop-off in architecture importance according to the descending order of accuracy", or "Rel@K computes the ratio of the accuracy of architecture $A_K$ to that of the best one $A_{max}$". The authors should not shy away from using mathematical symbols and instead replace them with confusing textual descriptions --- at the very least, precise definitions should be available in the appendix.

**Experimental settings:** I still have the following concerns:

- For a fair comparison, the authors use the same performance predictor setting, including the same learning rate `lr` and weight decay `wd` (Appendix B.2). However, this is inherently unfair for comparing loss functions, as different `lr` and `wd` lead to different losses. In fact, different losses are sensitive to `lr` and `wd`. For example, in information retrieval practices, pairwise loss typically requires a lower `lr` and `wd`, while listwise loss needs a higher `lr`. The authors should compare the loss functions across a wider range of hyperparameters and provide a sensitivity analysis to ensure a fair and comprehensive comparison.

- The authors test only on one performance predictor composed of a four-layer GCN encoder and a three-layer MLP, which is somewhat limited. I recommend that the authors conduct experiments on more types of performance predictors to verify the consistent performance of the loss functions across different networks.

**Metrics:** The authors introduce various metrics to evaluate performance, emphasizing that Top-$K$ metrics are more effective for practical NAS tasks. However, there are additional Top-$K$ ranking metrics in recommender systems which need to be considered:

- NDCG and NDCG@$K$ are the most commonly used metrics in information retrieval and recommendation systems. Many ranking loss functions are designed based on them, which are fundamentally different from the accuracy-based metrics listed in the paper. In fact, with slight modifications, NDCG can be adapted for evaluation in NAS. Specifically, by sorting architecture-performance pairs $(x_i, y_i)$ according to the predicted performance $\hat{y} _i$, DCG can be defined as $\mathrm{DCG} = \sum _{i = 1}^{N} (2^{y _i} - 1) / \log _2(i + 1)$ . I suggest the authors consider more recommendation metrics to evaluate ranking loss functions.

**Experiment Analysis:** The experimental analysis is generally thorough, but I have the following additional questions:

- In Section 4.1.1, the authors compare the effects of different loss functions on various NAS tasks and "observe that no ranking loss functions consistently surpass other competitors across all 13 tasks. For instance, ListNet achieves the top-1 $\tau$ in NAS-Bench-101 while having the lowest $\tau$ in the TransNAS-Bench101-Micro Autoencoder task". Why does this occur? Is it related to the dataset or task? A more insightful discussion is preferred.

- I suggest the authors summarize the criteria for choosing ranking loss functions after the experiments. Specifically, which type of loss function should be selected for a particular dataset size, NAS task, and training portion?



Additionally, I have a few **minor concerns** that do not impact the score:

- All instances of @K should be written as @$K$ for consistency.
- Figure 1 should highlight the best results, perhaps using superscripts.
- The legend in Figure 2 obstructs the x-axis label "Training Portion".
- The caption for Figure 2 uses "under various settings", which is confusing. It could be changed to "under different training and test portions".

**Questions:**

See the "Weaknesses" section.

---

### Official Review · Reviewer_N7Nn · 2024-10-28

**Soundness:** 2
**Presentation:** 2
**Contribution:** 2
**Rating:** 3
**Confidence:** 4

**Summary:**

The paper evaluates different ranking loss functions in performance predictors for Neural Architecture Search (NAS) and also draws some insights.

**Strengths:**

1. The paper is easy to read.
2. The experiments are comprehensive.

**Weaknesses:**

The paper conducts comparative experiments and analyzes existing loss functions for assessing the performance of Neural Architecture Search (NAS). However, the overall innovation and contribution of the paper are limited, with no new contributions in terms of evaluation methods, conclusions drawn, or new methods derived from the evaluation results. The two insights obtained through experiments also lack persuasiveness. The first insight, the importance of ranking loss in performance predictors, is widely recognized. It is precisely because people recognize the importance of ranking loss for NAS that there has been continuous iteration and the proposal of various ranking losses. The second insight, that ranking loss with excellent performance on top-centered rank metrics can help find high-quality architectures, is also quite straightforward. Does this insight imply that top-centered rank metrics should be used in the design of NAS methods? If the conclusion relies solely on experimental evaluation, can it stand up? Is there any theoretical support?

I suggest that having a clear or more in-depth conclusion regarding the loss function would be more persuasive, such as what kind of model or predictor is suitable for what kind of ranking loss, or analyzing the mathematical principles of different loss functions to further propose what principles we should follow when designing ranking loss functions.

Overall, I believe this paper does not make any special contributions in terms of experimental setup, conclusions drawn, and method design, and I think it does not meet the standards of ICLR.

**Questions:**

1. How can we utilize the conclusion "ranking loss with excellent performance on top-centered rank metrics can help find architectures with high quality" to guide the future design of NAS methods or the design of loss functions?
2. Can you explain the obtained insights from the mathematical essence of different loss functions?

---

### Official Review · Reviewer_ohRH · 2024-10-31

**Soundness:** 3
**Presentation:** 3
**Contribution:** 2
**Rating:** 5
**Confidence:** 2

**Summary:**

Disclaimer: I have never worked on NAS, not sure why this paper was assigned to me. Providing a high-level, low confidence review.

Overview:
This paper studies ranking losses for training predictors used in neural architecture search (NAS). Specifically, a search algorithm uses a predictor to evaluate candidate architectures since proper evaluation is often very expensive. Several ranking losses are compared, including pointwise, pairwise, and listwise losses. The paper argues that using weighted losses, which place more weight on top-ranking architectures, as opposed to simply ranking them overall, yields better performance than other losses.

A thorough comparison of several ranking losses for NAS may be an interesting contribution, but there could be some concerns regarding novelty.

**Strengths:**

* The comparison of multiple ranking losses seems comprehensive, covering 11 losses.
* Some of the proposed weighted losses show promising NAS results.
* The paper is clearly written.

**Weaknesses:**

* I am not an expert on NAS but adding a weighted ranking loss to previously proposed ranking losses may be somewhat Incremental (weighted vs. non-weighted), especially since improvement in performance compared to baselines seems rather small.
* The results are sometimes hard to interpret. For example, looking at Figure 1, it is hard to say if there is a loss which performs well across multiple tasks. Perhaps try a bar plot or a line plot? As another example, figures 2 and 4 show the winning loss for a combination of train portion, test portion, and task, and it is hard to identify clear trends in the multitude of results.

**Questions:**

Suggestion: move the Loss Function in Table 4 to the second column from the left. Perhaps also move “Search Method” to the third column.

---

### Official Review · Reviewer_YugU · 2024-11-01

**Soundness:** 2
**Presentation:** 1
**Contribution:** 1
**Rating:** 3
**Confidence:** 3

**Summary:**

The paper investigates the effectiveness of various ranking loss functions in performance predictors within Neural Architecture Search (NAS). In specific, this paper compares 11 ranking loss functions, including pointwise, pairwise, listwise, and weighted categories, across multiple search spaces and metrics to identify the most effective for NAS. The study finds that ranking loss choice significantly impacts predictor performance, particularly in discovering high-quality architectures. The paper finds that the top-centered metrics are better suited for NAS tasks than traditional metrics, emphasizing the predictor's ability to identify high-performing architectures. These findings help guide the selection of ranking losses, improving the efficiency and accuracy of predictor-based NAS methods.

**Strengths:**

1. Comprehensive work with extensive experiments.

**Weaknesses:**

Overall, I personally find it’s hard to justify the potential impact of the work. This is not the first work studying about ranking losses in efficient neural architecture search or autoML in general. Ranking losses haven’t been widely used in NAS or autoML likely because still lack of significant and consistent gain from ranking losses in practice.

In addition, I found the following points making the paper hard to read and understand by general audience.
1. Mathematical definitions of both losses and metrics are missing, not even in appendix. I had to refer to other papers. Without math definitions, details of the metrics are hard to understand. For example, N@K is the lower the better, which is only mentioned in the caption of Figure 4, likely to confuse many readers at the beginning.
2. Color code of the results are confusing. For example, the color code in  Figure 1 appears to highlight the very bad ones, like MSE on TB101_MACRO-AUTO dataset. However, I believe what’s more relevant is the best loss on each dataset. By just scanning, hard to see any pattern showing ranking losses superior on NAS.
3. Figure captions are not very informative. Important explanations are missing. For example, by just looking at Figure 2 and 4 and their captions, almost impossible to understand why colors are shown on Train-test portion grids. And thus hard to get what these colors are trying to tell.

**Questions:**

1. What is the scale of the N@k metric? My read of the definition is it’s the “true rank” of the architecture then it relies on the context, what are the architectures ranked together with the one on list and what is the size of the context?

---

### Note · Authors · 2024-11-13

I have read and agree with the venue's withdrawal policy on behalf of myself and my co-authors.